# Only Subclinical Alterations in the Haemostatic System of People with Diabetes after COVID-19 Vaccination

**DOI:** 10.3390/v15010010

**Published:** 2022-12-20

**Authors:** Margret Paar, Faisal Aziz, Caren Sourij, Norbert J. Tripolt, Harald Kojzar, Alexander Müller, Peter Pferschy, Anna Obermayer, Tamara Banfic, Bruno Di Geronimo Quintero, Nandu Goswami, Axel Schlagenhauf, Martin Köstenberger, Thomas Bärnthaler, Thomas Wagner, Andelko Hrzenjak, Willibald Wonisch, Gilbert Reibnegger, Reinhard B. Raggam, Harald Sourij, Gerhard Cvirn

**Affiliations:** 1Division of Medicinal Chemistry, Otto Loewi Research Centre for Vascular Biology, Immunology and Inflammation, Medical University of Graz, 8010 Graz, Austria; 2Division of Endocrinology and Diabetology, Interdisciplinary Metabolic Medicine Trials Unit, Medical University of Graz, 8010 Graz, Austria; 3Division of Cardiology, Medical University of Graz, 8010 Graz, Austria; 4Division of Physiology, Otto Loewi Research Centre for Vascular Biology, Immunology and Inflammation, Medical University of Graz, 8010 Graz, Austria; 5Department of Pediatrics and Adolescent Medicine, Division of General Pediatrics, Medical University of Graz, 8010 Graz, Austria; 6Division of Pharmacology, Otto Loewi Research Centre for Vascular Biology, Immunology and Inflammation, Medical University of Graz, 8010 Graz, Austria; 7Department of Blood Group Serology and Transfusion Medicine, Medical University of Graz, 8010 Graz, Austria; 8Division of Pulmonology, Department of Internal Medicine, Medical University of Graz, 8010 Graz, Austria; 9Division of Angiology, Medical University of Graz, 8010 Graz, Austria

**Keywords:** COVID-19, type 1 diabetes, type 2 diabetes, platelet function, thrombin generation, thrombelastometry

## Abstract

People with diabetes have an increased risk of experiencing adverse COVID-19 outcomes. COVID-19 vaccination is, therefore, highly recommended. However, people with diabetes have an inherently elevated risk of thrombotic events and the impact of the vaccination on the coagulation system in this patient population remains to be elucidated. The aim of this study was to investigate the impact of COVID-19 vaccination on the haemostatic system in people with type 1 or type 2 diabetes. We evaluated the effects of COVID-19 vaccination (BioNTech Pfizer, Moderna, AstraZeneca) on standard coagulation parameters, whole blood coagulation (Thrombelastometry), platelet function (impedance aggregation), and thrombin generation (calibrated automated thrombography) in people with type 1 diabetes mellitus (*n* = 41) and type 2 diabetes mellitus (*n* = 37). Blood sampling points were prior to vaccination and two weeks after the respective vaccination. Thrombelastometry measurements indicated moderately increased clot formation post-vaccination in people with type 1, as well as with type 2, diabetes: “Clot formation times” were significantly shorter, and both “maximum clot firmness” and “alpha angles” were significantly higher, as compared to the respective pre-vaccination values. Therefore, TEM parameters were not altered after vaccination in patients receiving ASA. Moreover, platelet aggregation was enhanced in people with type 1 diabetes, and plasma levels of D-Dimer were increased in people with type 2 diabetes, following COVID-19 vaccination. All other standard coagulation parameters, as well as thrombin generation, were not affected by the vaccination. The coagulation responses of people with diabetes to COVID-19 vaccination were only subclinical and comparable to those observed in healthy individuals. Our findings suggest that people with diabetes do not face an increased activation of the coagulation post-vaccination.

## 1. Introduction

The coronavirus disease-19 (COVID-19) pandemic, caused by the severe acute respiratory syndrome coronavirus 2 (SARS-CoV-2), poses a huge health threat to the population and has resulted in significant disruptions to healthcare and social systems globally [1]. Vaccination against COVID-19 is the most promising chance to control this pandemic [2].

Given the large number of vaccinations, very rare, but serious, adverse events have been recorded. Application of both adenovirus-based vaccines, as well as mRNA vaccines, has been shown to be associated with enhanced inflammation and coagulation activation in healthy individuals [3]. As a consequence, rare adverse events like vaccine-induced immune thrombocytopenia and thrombosis (VITT) [4] and cerebral venous sinus thrombosis (CVST) [5] have been reported following COVID-19 vaccination with mRNA vaccines, and are more pronounced with adenovirus-based vaccines [6].

While the benefit–risk profile of these COVID-19 vaccines is clearly in favour of the vaccination, the impact on the coagulation system, particularly in people already at increased risk of atherothrombotic events, such as those with manifest diabetes mellitus, deserves further investigation [7].

We, therefore, investigated the impact of COVID-19 vaccination (BioNTech Pfizer, Moderna, and AstraZeneca) on the haemostatic system in people with type 1 and type 2 diabetes.

## 2. Materials and Methods

### 2.1. Subjects and Experimental Design

The present study is part of the ‘Immune response to COVID-19 vaccination in people with Diabetes Mellitus—COVAC-DM’ study, described in detail previously [8]. In brief, adults with type 1 or type 2 diabetes, aged 18 to 80 years, who were diagnosed with diabetes prior to receiving a COVID-19 vaccine and who were willing to give written informed consent, were included in the study. Of the original 161 individuals, a complete set of coagulation data was available from 78 individuals and is thus presented herein. Individuals with clotting disorders or receiving oral anticoagulants were excluded. As the vaccination was performed within the Austrian COVID-19 vaccination programme, participants received adenovirus- and mRNA-based vaccines, with a clear preponderance of the latter.

A physical examination was performed and, for the present analysis, 9 mL of venous blood was collected in pre-citrated Vacuette^®^ tubes, which contained 3.8% sodium citrate (Greiner Bio-one GmbH, Kremsmünster, Austria). Blood samples were collected 60 to 2 days prior to the first vaccination (baseline values), 7 to 14 days after the first vaccination (visit 1) and 14 to 21 days after the second vaccination (visit 2), see Figure 1. Haemoglobin, platelet counts, thrombelastometry (TEM), and platelet aggregation were measured in citrated whole blood (WB) samples. Subsequently, the remaining whole blood underwent centrifugation at 500 g for 20 min in order to prepare platelet poor plasma (PPP) samples. The remaining measurements were performed in PPP samples. The study protocol was approved by the ethics committee of the Medical University of Graz (33-366 ex 20/21). The study was conducted according to the guidelines of Good Clinical Practice and the Declaration of Helsinki. Participants were informed about all study procedures by a physician and provided written informed consent.

Seventy-eight individuals with diabetes (41 with type 1, 37 with type 2) were included in this prospective substudy of the COVAC-DM study.

The effects of COVID-19 vaccination on standard coagulation values, whole blood coagulation (thrombelastometry, TEM, Matel Medizintechnik, Graz, Austria), platelet aggregation (impedance aggregometry, Probe and Go, Endingen, Germany), and thrombin generation (calibrated automated thrombography, CAT, Thrombinoscope BV, Maastricht, The Netherlands) were evaluated prior to vaccination, after the first, and after the second vaccination.

### 2.2. Standard Coagulation Markers

Determinations of activated partial thromboplastin times (APTTs), prothrombin times (PTs), and plasma levels of fibrinogen were performed on a Atellica COAG 360 Coagulation System (Siemens Healthcare Diagnostics GmbH, Vienna, Austria). Human D-Dimer ELISA Kit was purchased from Sigma-Aldrich Handels GmbH (Vienna, Austria). Haemoglobin and platelet counts were determined on a Sysmex KX-21 N Automated Haematology Analyzer from Sysmex (Lincolnshire, IL, USA).

### 2.3. Whole Blood Tissue Factor-Triggered TEM Assay

The time course of clot formation was measured using the TEM coagulation analyzer (ROTEM^®^05) from Matel Medizintechnik (Graz, Austria), leading to four parameters: “Coagulation time” (CT), the time from adding the trigger until initial fibrin formation; “Clot formation time” (CFT), the time until the amplitude reaches 20 mm; “Maximum clot firmness” (MCF), reflecting clot stability; and the “alpha angle”, indicating the velocity of fibrin built-up and cross-linking. The final sample volume was 340 μL. Clot formation was triggered with the addition of 40 μL of “trigger solution” (containing 0.35 pmol/L recombinant human tissue factor (TF) and 3 mmol/L CaCl_2_, final concentration) to 300 μL of citrated WB. Sorensen et al. have described this method previously [9]. TF thromboplastin (Innovin^®^) was purchased from Dade Behring Marburg GmbH (Marburg, Germany). The lyophilized product was dissolved in distilled water (10 mL) and then diluted at a ratio of 1:1000 in 0.9% sodium chloride solution from Fresenius Kabi Austria GmbH (TF-stock solution).

### 2.4. Impedance Aggregation Assay

The Chrono-Log Aggregometer Model 590 from Probe and Go (Endingen, Germany) was used to assess platelet aggregation. This device is based on the impedance method [10]. Results are presented as amplitude (or maximum aggregation) in Ohm at six minutes after reagent addition and as lag time (or aggregation time) in seconds, the time interval until the onset of platelet aggregation. The “Slope” reflects the rate of platelet aggregation and is expressed in Ohm/min. Collagen (2 μg/mL final concentration), purchased from Probe and Go (Endingen, Germany), was used as a trigger of aggregation, as previously described [11].

### 2.5. Automated Fluorogenic Measurement of Thrombin Generation

Calibrated automated thrombography (CAT, Thrombinoscope BV, Maastricht, The Netherlands) was used to monitor thrombin generation curves [12]. The capability of a given plasma sample to form thrombin was monitored with respect to lag time preceding the thrombin burst (lag Time), time to peak, peak height (Peak), maximum velocity of thrombin formation (VelIndex) and endogenous thrombin potential (ETP), and the time point of free thrombin disappearance (StartTail). Measurements were performed in the presence of 5 pM of TF (final concentration). Thrombin generation wavelengths were: 390 nm (excitation) and 460 nm (emission). Measuring the formation of thrombin by using CAT is an appropriate method to assess the coagulability of plasma samples [13]. Experiments were performed in a random sample of 10 (out of 41) participants with type 1 diabetes (T1DM) and in 10 participants (out of 37) with type 2 diabetes (T2DM).

### 2.6. Statistics

The GraphPad 8.0 Prism package was used for statistical evaluation. Differences in Table 1 between participants with type 1 diabetes mellitus (T1DM) and participants with type 2 diabetes mellitus (T2DM) were determined by means of Student’s *t* test (quantitative variables) or by means of Fischer’s exact test (qualitative variables). The significance of the influence of vaccination on coagulation variables was calculated by means of Friedman test and Dunn’s multiple comparison test in T1DM participants (Table 2), as well as in T2DM participants (Table 3). All *p* values of ≤0.05 were considered statistically significant. * *p* ≤ 0.05, ** *p* ≤ 0.01, *** *p* ≤ 0.001.

## 3. Results

Patients’ characteristics (anthropometric values, comorbidities, microvascular complications, and concomitant therapies) are listed in Table 1. Forty-one participants had T1DM and 37 participants had T2DM. Unsurprisingly, participants with T2DM were significantly older than participants with T1DM (*p* < 0.001). Consistently, participants with T2DM had significantly higher BMIs and a higher prevalence of hypertension, liver disease, as well as of polyneuropathies. Of all participants included in this analysis, 87.2% received the BioNTech Pfizer, 6.4% the Moderna, and 6.4% the AstraZeneca vaccine. Vaccine distribution was similar in people with type 1 and type 2 diabetes (*p* = 1.000). The duration of diabetes was significantly shorter for participants with T2DM (*p* < 0.001). Distribution of well-controlled (HbA1c < 58 mmol/mol) and insufficiently controlled (HbA1c > 58 mmol/mol) participants was similar in both the T1DM and T2DM groups (*p* = 0.176). Significantly more participants with type 2 diabetes received acetylsalicylic acid (ASA, *p* = 0.020). One woman was on oral contraceptive treatment.

### 3.1. Impact of COVID-19 Vaccination on the Haemostatic System of Participants with Type 1 Diabetes

Standard coagulation markers were not affected by the COVID-19 vaccination in this group of participants except that PTs were slightly, but significantly, prolonged after the first and the second vaccination compared to baseline values (*p* = 0.020), listed in Table 2. Plasma levels of D-Dimer were numerically increased after vaccination albeit not reaching statistical significance (*p* = 0.052). 

TEM measurements indicated a vaccination-induced coagulation activation: CFTs were significantly shortened (*p* < 0.001), and both MCF values and alpha angles were higher compared to the respective baseline values (*p* < 0.001, Table 2).

Impedance aggregometry measurements indicated a vaccination-induced increase in platelet function, listed in Table 2. Amplitudes and slopes were higher (*p* < 0.001 and *p* = 0.019, respectively), and lag times were significantly shortened (*p* < 0.001) after the second vaccination compared with the respective baseline values. Impedance aggregometry values of the three patients receiving ASA were similar to those of the patients not receiving ASA.

CAT measurements did not indicate any influence of the COVID-19 vaccination on the ability of the respective plasma samples to generate thrombin, listed in Table 2.

### 3.2. Impact of COVID-19 Vaccination on the Haemostatic System of Participants with Type 2 Diabetes

Plasma levels of D-Dimer were slightly altered but within the normal range after the second vaccination (*p* = 0.001), and all other standard coagulation markers were not affected by vaccination, as listed in Table 3. 

TEM measurements indicated, similar to participants with T1DM, a coagulation activation after vaccination: CFTs were significantly shortened (*p* = 0.034), and both MCF values and alpha angles were higher compared to the respective baseline values (*p* = 0.001 and *p* = 0.049, respectively, Table 3). Since 29.7% of the participants with T2DM received ASA, we compared the influence of vaccination on TEM values in participants with T2DM receiving ASA with that in participants not receiving ASA. In patients not receiving ASA, CFTs were significantly shortened (*p* = 0.019) and both MCF values and alpha angles were higher compared to baseline levels (*p* = 0.001 and *p* = 0.0316, respectively). However, TEM values were not affected by vaccination in participants receiving ASA. CFTs (*p* = 0.219), MCFs (*p* = 0.629), and alpha angles (*p* = 0.219) were approximately the same at baseline and after vaccination. Apparently, taking ASA masks the shortening of CFTs and the increases of MCF and alpha angle as observed in patients without ASA.

In contrast to the T1DM group, no increased platelet function after vaccination was observed. Amplitudes, slopes, and lag times were virtually the same prior and after vaccination (*p* = 0.593, *p* = 0.433, and *p* = 0.206, respectively), as listed in Table 3. This was also true if we only analysed T2DM participants not taking ASA. Amplitudes, slopes, and lag times were virtually the same prior and after vaccination (*p* = 0.582, *p* = 0.456, and *p* = 0.224, respectively) in this sub-group.

Similar to the T1DM group, CAT measurements were not indicative of any influence of the COVID-19 vaccination on the ability of the respective plasma samples to generate thrombin, listed in Table 3.

### 3.3. Impact of COVID-19 Vaccination on the Haemostatic System with Respect to Glycemic Control

Vaccination-induced effects on the haemostatic system were virtually the same in well-controlled patients (HbA1c < 58 mmol/mol) as in insufficiently controlled patients (HbA1c > 58 mmol/mol), with one exception. TEM-derived CTs were significantly shorted by the vaccination in insufficiently controlled T1DM patients from 253 [202–284] s at baseline to 201 [191–276] s at visit1 (*p* = 0.046). TEM-derived CTs were not significantly affected by the vaccination in well-controlled T1DM patients (*p* = 0.291).

In the overall study, we did not observe any cases of deep venous thrombosis, pulmonary artery embolism, or sinus venous thrombosis up to three weeks following the second vaccination.

## 4. Discussion

This study demonstrated a moderate coagulation activation in response to COVID-19 vaccination in people with type 1 and type 2 diabetes. We found vaccination-induced increased clot formation, as determined by TEM measurements as well as elevated plasma levels of D-Dimer, in people with type 2 diabetes. Moreover, platelet aggregation was increased after vaccination in people with type 1 diabetes. Most of the alterations were within the normal range of the respective parameter.

Since people with diabetes are at a high risk of experiencing adverse COVID-19 outcomes, COVID-19 vaccination is highly recommended in this patient population [15,16,17].

However, potential adverse events of a vaccine should be taken seriously and hence a potential activation of the coagulation system should be considered [18]. This is of particular interest, as people with diabetes are already at increased risk of thromboembolic events. [7,19,20]

It was therefore the aim of our study to evaluate possible coagulation activation after COVID-19 vaccination in people with type 1 or type 2 diabetes. We actually found a partial activation of the coagulation system. The main findings were an increased readiness to form clots after vaccination (evaluated by means of TEM) in blood from patients with type 1 and type 2 diabetes as well as a vaccine-induced higher platelet aggregation in participants with type 1 diabetes. D-Dimer was elevated post-vaccination in participants with type 2 diabetes. It has to be stated that these changes, although statistically significant, were rather small, with very limited clinical relevance [14].

Interestingly, platelet aggregation in participants with type 2 diabetes, unlike in participants with type 1 diabetes, was not affected by the COVID-19 vaccination. This is likely due to the higher intake of ASA in this patient population (*p* = 0.020 vs. the T1DM sub-group). It is well known that ASA is a potent inhibitor of platelet aggregation [21,22]. However, we did not find vaccine-induced alterations in platelet aggregation, even in a sub-group of participants with type 2 diabetes who were not receiving ASA.

Since the incidence of thrombosis has been shown to increase with increasing age, it is generally assumed that the haemostatic system of older individuals is hypercoagulable when compared to that of the young [23]. The findings of our study do not support this assumption with respect to coagulation activation after COVID-19 vaccination. Although the T2DM patients in our study were significantly older than the T1DM patients, the coagulation activation after vaccination was not more pronounced in our T2DM patients. On the contrary, while both TEM and aggregometry values were elevated after vaccination in T1DM patients, only TEM values were elevated in the T2DM patients not receiving ASA. In T2DM patients receiving ASA, TEM values were not altered after vaccination. Presumably, the increased thrombosis propensity in the elderly is not primarily attributable to a hyperactive coagulation system but to other pathologies associated with aging, including malignant disease or major surgery [24].

Diabetes mellitus confers about a two-fold excess risk of a wide range of vascular diseases associated with coagulation activation, e.g., stroke and myocardial infarction [25,26]. It therefore stands to reason that coagulation activation after COVID-19 vaccination is more pronounced in people with diabetes. However, our results do not support this assumption. We found virtually the same (moderate) coagulation activation after vaccination in well-controlled patients (HbA1c < 58 mmol/mol) and in insufficiently controlled patients (HbA1c > 58 mmol/mol). This could be due to the low number of patients in our study. 

The moderate coagulation activation after vaccination in people with diabetes in the present study is comparable to that in healthy individuals, as shown in a recent study by Ostrowski et al. [3]. They have shown that in healthy individuals, TEM-derived MCFs and platelet aggregation markers were increased post-vaccination to approximately the same degree as in our analysis. Another common feature of the two studies is that COVID-19 vaccination did not alter the capability of plasma samples to generate thrombin (the “Thrombin potential”). Unfortunately, both studies need to be compared with caution. Ostrowski et al. did not derive their pre-vaccination samples from the same individuals that were used to assess the effects of vaccination on the coagulation system. Moreover, they used the Multiplate^®^ device to perform platelet aggregation measurements while we used the Chrono-Log aggregometer^®^, and finally we used significantly lower amounts of TF to trigger clot formation in the TEM experiments. This low level trigger has been shown to allow sensitive evaluation of influential factors on the coagulation system [9,27].

Participants in our study have received two doses of the same vaccine. It has been shown that a heterologous vaccination schedule could induce a significantly higher humoral response in people with diabetes [28]. However, data on coagulation were not reported in this manuscript. Hence, we cannot exclude that a heterologous vaccination scheme might have a different impact on coagulation.

In summary, we state that the coagulation responses of diabetic patients to COVID-19 vaccination are moderate and are comparable to those of healthy individuals and did not translate into any clinical thromboembolic events in our cohort. 

A limitation of our study is the low number of participants, and, moreover, that we did not analyse for mRNA vaccines (BioNTech Pfizer and Moderna) and adenoviral vaccine from AstraZeneca separately due to the sample size. The coagulation response to adenoviral vaccines is apparently more pronounced than that to mRNA vaccines [3]. It has been speculated that specific components of the adenovirus vector may serve as initial triggers of platelet activation and thrombin generation and thus promote the development of VITT or CVST when combined with high-titer, functionally active platelet factor 4 (PF4) antibodies [3,29]. However, we performed our study within the setting of the national vaccination strategy of Austria. Therefore, the distribution of vaccines in our study represents the actual distribution in Austria. Due to the low number of non-Pfizer vaccine recipients (5 for both Moderna and AstraZeneca) a sub-group analysis was not feasible.

A further limitation of the study is that the vaccination-associated changes in several coagulation parameters were, although significant from a statistical/mathematical point of view, small and without noted clinical significance. Thus, individuals’ own biological fluctuations, together with laboratory variability, might have influenced the data presented.

COVID-19 vaccination had no impact on the haemostic system of patients with polyneuropathy. However, in people without polyneuropathy, D-Dimer levels were significantly increased, CTs were significantly shortened, MCF and alpha angles were significantly higher and platelet activation was slightly, but significantly, enhanced after vaccination. To our knowledge, the underlying mechanisms explaining this relative insusceptibility of people with polyneuropathy are not known today and should be clarified in a future study.

Importantly, our results indicate that diabetic patients are not at a higher risk of COVID-19 vaccination-induced coagulation activation than healthy individuals. Thus, we do not expect more thrombotic events to be triggered more frequently by COVID-19 vaccination in people with diabetes than in healthy people.

The data presented herein support the safety profile of COVID-19 vaccination in people with diabetes, a population in which the vaccination is highly recommended [30].

## 5. Conclusions

The coagulation responses of people with diabetes to COVID-19 vaccination were only subclinical and comparable to those observed in healthy individuals. Our findings suggest that people with diabetes do not face an increased activation of coagulation post-vaccination.

## Figures and Tables

**Figure 1 viruses-15-00010-f001:**
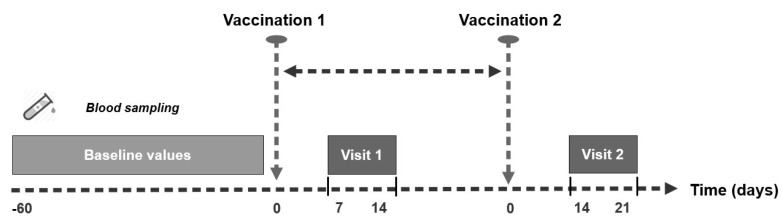
Vaccination administration schedule according to the Austrian Vaccination Policy.

**Table 1 viruses-15-00010-t001:** Patients’ characteristics. Qualitative variables are presented as frequencies and percentages (%). Quantitative variables are presented as means and standard deviations (±SD). Fischer’s exact test was applied to compare qualitative variables with diabetes groups. Student’s *t* test was applied to compare quantitative variables with diabetes groups. Abbreviations: ADP, adenosine diphosphate; ASA, acetylsalicylic acid; CABG, coronary artery bypass graft; BMI, body mass index; HbA1c, glycated haemoglobin; OA, oral anticoagulants; PTCA, percutaneous transluminal coronary angiography; TIA, transient ischemic attack; T1DM, type 1 diabetes; T2DM, type 2 diabetes.

Variables	All (*n* = 78)	T1DM (*n* = 41)	T2DM (*n* = 37)	T1DM vs. T2DM *p* Value
Age, years	50.5 (12.5)	44.6 (13.7)	57.1 (6.5)	**<0.001**
Female sex, *n* (%)	30 (38.5)	19 (46.3)	11 (29.7)	0.170
BMI, kg/m^2^	28.3 (5.6)	25.3 (4.7)	31.6 (4.4)	**<0.001**
Smokers, *n* (%)	35 (44.9)	22 (53.7)	13 (35.1)	0.270
Vaccine, *n* (%) BioNTech Pfizer Moderna AstraZeneca	68 (87.2) 5 (6.4) 5 (6.4)	36 (87.8) 3 (7.3) 2 (4.9)	32 (86.5) 2 (5.4) 3 (8.1)	1.000
Duration of diabetes, years	17.9 (13.7)	22.9 (15.4)	12.3 (8.9)	**<0.001**
HbA1c > 58 mmol/mol	30 (38.5)	13 (31.7)	17 (45.9)	0.176
Comorbidity Hypertension, *n* (%) Coronary heart disease, *n* (%) Myocardial infarction, *n* (%) TIA, *n* (%) Heart failure, *n* (%) PTCA/CABG, *n* (%) Stroke, *n* (%) Liver disease, *n* (%) History of cancer, *n* (%) Microvascular complications Retinopathy, *n* (%) Polyneuropathy, *n* (%) Concomitant therapy ASA, *n* (%) ADP, *n* (%)	35 (44.9) 4 (5.1) 1 (1.2) 3 (3.8) 0 (0.0) 4 (5.0) 1 (1.3) 11 (14.1) 4 (5.0) 14 (17.9) 17 (21.8) 14 (17.9) 1 (1.3)	10 (24.4) 1 (2.4) 0 (0.0) 0 (0.0) 0 [0.0) 0 (0.0) 0 (0.0) 0 (0.0) 1 (2.4) 9 (22.0) 4 (9.8) 3 (7.3) 0 (0.0)	25 (67.6) 3 (8.1) 1 (2.5) 3 (8.1) 0 (0.0) 4 (10) 1 (2.7) 11 (29.7) 3 (8.1) 5 (13.5) 13 (35.1) 11 (29.7) 1 (2.7)	**<0.001** 0.057 0.160 0.055 0.120 0.230 0.240 **<0.001** 0.360 0.570 **0.004** **0.020** 0.490

**Table 2 viruses-15-00010-t002:** Coagulation markers pre- and post-vaccination in T1DM patients. Data are expressed as median and IQR. *p* Values were calculated by means of the Friedman test (*n* = 41). Abbreviations: APTT, activated partial thromboplastin time; PT, prothrombin time. Coagulation markers pre- and post-vaccination in T1DM patients. Data are expressed as median and IQR. *p* Values were calculated by means of the Friedman test (*n* = 41). Abbreviations: APTT, activated partial thromboplastin time; PT, prothrombin time. ^a^ see references [11,14]; ^b^ reference values do not exist.

	Baseline Median [IQR]	Visit 1 Median [IQR]	Visit 2 Median [IQR]	*p* Value (Friedman)	Reference Values ^a^
CRP [mg/L] Standard coagulation markers APTT [s] PT [%] D-Dimer [µg/mL] Fibrinogen [mg/dL] Haemoglobin [g/dL] Platelets [10^6^/µL]	2 [1–4] 32 [30–34] 111 [102–120] 0.36 [0.25–0.57] 265 [231–321] 14 [13–15] 264 [204–296]	1 [1–2] 32 [30–34] 116 [104–120] 0.40 [0.26–0.56] 270 [231–312] 14 [13–15] 261 [223–310]	1.5 [1–3] 32 [30–34] 116 [105–120] 0.51 [0.28–0.58] 264 [244–299] 14 [13–15] 263 [216–303]	0.134 0.177 **0.020** 0.052 0.601 0.166 0.495	<3 30–40 70–130 0–0.5 180–350 13 [12–15] 250 [125–318]
Thrombelastometry Coagulation time [s] Clot formation time [s] Maximum clot firmness [mm] Alpha angle [°]	227 [196–258] 193 [152–242] 55 [50–61] 57 [52–62]	214 [189–239] 130 [107–157] 61 [57–65] 64 [58–69]	223 [210–248] 145 [116–175] 61 [56–64] 62 [57–67]	0.104 **<0.001** **<0.001** **<0.001**	256 ± 25 139 ± 40 58 ± 5 64 ± 7
Platelet aggregation Amplitude [Ohm] Slope [Ohm/min] Lag time [s]	12 [11–14] 7 [6–9] 81 [63–103]	11 [10–12] 7 [5–9] 89 [70–110]	13 [10–14] 8 [6–10] 70 [57–84]	**<0.001** **0.019** **<0.001**	13.6 ± 1.4 8.4 ± 2.0 58.6 ± 9.7
Thrombin generation Lag time [min] Thrombin potential [nM·min] Peak [nM] Time to Peak [min] VelIndex [nM/min] StartTail [min]	2.7 [2.3–2.9] 1449 [1180–1573] 192 [157–233] 7.3 [6.2–8.0] 42 [32–60] 25 [23–27]	2.7 [2.5–3.1] 1446 [1302–1690] 199 [131–231] 7.9 [6.7–9.1] 41 [22–56] 25 [24–30]	2.7 [2.3–3.0] 1407 [1310–1640] 178 [159–231] 7.3 [6.7–8.6] 39 [29–56] 25 [23–27]	0.161 0.779 0.558 0.098 0.205 0.205	- ^b^ - ^b^ - ^b^ - ^b^ - ^b^ - ^b^

**Table 3 viruses-15-00010-t003:** Coagulation markers pre- and post-vaccination in T2DM patients. Data are expressed as median and IQR. *p* Values were calculated by means of the Friedman test (*n* = 37). Abbreviations: APTT, activated partial thromboplastin time; PT, prothrombin time. ^a^ see references [11,14]; ^b^ reference values do not exist.

	Baseline Median [IQR]	Visit 1 Median [IQR]	Visit 2 Median [IQR]	*p* Value (Friedman)	Reference Values ^a^
CRP [mg/L] Standard coagulation markers APTT [s] PT [%] D-Dimer [µg/mL] Fibrinogen [mg/dL] Haemoglobin [g/dL] Platelets [10^6^/µL]	2 [1–4] 31 [29–34] 120 [112–120] 0.32 [0.26–0.41] 288 [264–338] 15 [14–16] 238 [207–284]	2 [1–3] 32 [29–34] 120 [113–120] 0.32 [0.27–0.46] 295 [270–340] 15 [14–16] 257 [215–309]	2 [1–3] 31 [29–34] 120 [116–120] 0.40 [0.29–0.56] 304 [270–337] 15 [14–16] 243 [208–283]	0.824 0.530 0.100 **0.001** 0.342 0.766 0.061	<3 30–40 70–130 0–0.5 180–350 13 [12–15] 250 [125–318]
Thrombelastometry Coagulation time [s] Clot formation time [s] Maximum clot firmness [mm] Alpha angle [°]	240 [220–285] 181 [134–252] 56 [51–61] 58 [50–64]	240 [211–263] 148 [128–181] 61 [58–65] 61 [58–64]	247 [223–293] 154 [135–193] 60 [56–64] 60 [56–65]	0.241 **0.034** **0.001** **0.049**	256 ± 25 139 ± 40 58 ± 5 64 ± 7
Platelet aggregation Amplitude [Ohm] Slope [Ohm/min] Lag time [s]	11 [9–14] 6 [4–8] 91 [76–119]	12 [9–14] 7 [5–8] 93 [83–130]	12 [9–13] 7 [5–9] 90 [71–115]	0.593 0.433 0.206	13.6 ± 1.4 8.4 ± 2.0 58.6 ± 9.7
Thrombin generation Lag time [min] Thrombin potential [nM·min] Peak [nM] Time to Peak [min] VelIndex [nM/min] StartTail [min]	3.1 [2.9–4.0] 1388 [1303–1553] 182 [156–229] 8.2 [6.6–9.0] 40 [29–62] 25 [23–28]	3.2 [3.0–3.6] 1567 [1292–1664] 190 [172–225] 8.1 [6.8–8.5] 44 [32–56] 25 [24–26]	3.1 [2.8–3.6] 1381 [1173–1502] 182 [144–221] 7.7 [7.2–8.2] 40 [28–56] 25 [23–26]	0.223 0.067 0.316 0.974 0.601 0.368	- ^b^ - ^b^ - ^b^ - ^b^ - ^b^ - ^b^

## Data Availability

The authors hereby declare that the data presented in this study will be presented upon request by the corresponding author.

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
