# Peer review of "Only Subclinical Alterations in the Haemostatic System of People with Diabetes after COVID-19 Vaccination"

_viruses, 2022, doi:10.3390/v15010010_

Round 1

Reviewer 1 Report

In this manuscript, Cvirn et al investigated the effect of the COVID-19 vaccination on hemostasis in patients with DM, suggesting that the vaccination does not alter the hemostasis profoundly in these people. The study was properly executed, however, I have several comments 1. I suggest that you include a diagram detailing patient visits and patient disposition to make it more visual for the reader. Correct the inverted commas in the line 74.

2. I suggest indicating whether only patients with coagulation disorders and those treated with anticoagulants were excluded, or whether any of these patients were taking other drugs that might enhance a prothrombotic state (contraceptives, corticoids, for example) and were included in the study. 

3. Indicate at which wavelength thrombin generation was measured.

4. Correct the percentages in lines 160 and 161 of the text according to the percentages reflected in the vaccine types section of table 1. (87,6 vs 87.2, 6.2 vs 6.4) 

5. The authors performed a study in which they observed that treatment with ASA does not affect the results obtained in aggregometry. However, ROTEM parameters could also be affected and the authors do not comment on this. I propose to realise and include a statistical study comparing patients treated with ASA vs. not treated with ASA, to see if there is a significant difference between both groups of patients and if this treatment could be affecting the results obtained. Similarly, a separate study should be conducted to see if microvascular complications could be affecting the results of the study presented. 

6. The authors indicate that these patients received two doses of vaccination. Please specify in particular that both doses were of the same vaccine type. Do other trials review haemostatic testing of vaccinated DM patients  with homogenous or heterologus vaccination? If so, please include such a literature review in your discussion. If not, could your results translate to other vaccination schemes as well?  Please discuss.

7. A significant difference is observed in the values of PT, MCF, alpha angle and slope in patients with type 1 diabetes, as well as in the values of MCF and alpha angle in patients with type 2 diabetes, but which, nevertheless, remain within the normal values provided by the authors themselves. Reason whether this moderate coagulation activation is due to the effect of vaccination or whether it is due to the individual's own physiological fluctuations. 

8. It has been widely reported that there is a relationship between inflammatory states and blood clotting (include reference to this). For this reason, it would be interesting for the authors to assess the levels of some marker of vascular inflammation, such as PAI-1 and correlate it with the results obtained in ROTEM. 

9. Do the authors believe that the time of disease progression (longer in the subgroup of patients with type 1 diabetes) could be affecting the results obtained, thus observing more significant differences in this subgroup of patients? Please discuss 

10. I have some remarks regarding the thrombin generation assay which did not show any difference between visits. The authors evaluate the thrombin generation in PPP employing FT. Given that the WB samples were centrifuged only once to obtain PPP, microparticles could remained in the plasma. Have the authors measured any extracellular vesicle or microparticle content in these samples? 

11. Were baseline, V1 and V2 laboratory assays performed simultaneously or in different batches? If so, what is the effect of laboratory variability in your assays for the interpretation of results? 

Author Response

Response to Reviewer 1:

We thank the reviewer for the valuable comments helping to improve the quality of our manuscript!

Ad 1)

We have added now a diagram detailing patient visits, see Figure 1. We inverted the commas in the line 74.

Ad 2)

Patients taking anticoagulants and/or immunosuppressants were excluded from the study. Likewise, we excluded people with active malignancies (with the exception of intraepithelial neoplasia of the prostate gland and the gastrointestinal tract), those with acute inflammatory disease or immunosuppressant therapy. Women with oral contraception were not excluded.

Ad 3)

Thrombin generation wavelengths were: 390 nm (excitation), 460 nm (emmission).

Ad 4)

The percentages in lines 160 and 161 are corrected now.

Ad 5)

We now describe the possible effects of ASA on ROTEM parameters in chapter 3.2 (lines 221-228):

“Since 29.7% of the participants with T2DM received ASA, we compared the influence of vaccination on TEM values in participants with T2DM receiving ASA with that in partic-ipants not receiving ASA. In patients not receiving ASA, CFTs were significantly short-ened (p = 0.019) and both MCF values and alpha angles were higher as compared to base-line levels (p = 0.001 and p = 0.0316, respectively). However, TEM values were not affected by vaccination in participants receiving ASA. CFTs (p = 0.219), MCFs (p = 0.629), and al-pha angles (p = 0.219) were approximately the same at baseline and after vaccination.”

Additionally, it is stated now in the abstract:

“Whereby, TEM parameters were not altered after vaccination in patients receiving ASA.”, lines 44 and 45.

Additionally, it is stated now in the Discussion section:

“In T2DM patients receiving ASA, TEM values were not altered after vaccination.”, line 296.

We have added the new chapter 3.4 describing the influence of polyneuropathy (retinopathy was the same in both groups) on the results of the study (lines 249-260):

“3.4 Impact of COVID-19 vaccination on the haemostatic system of T2DM patients with and without microvascular complications

A high percentage of T2DM patients had polyneuropathy (35.1 %, Table 1). We, therefore, compared the impact of COVID-19 vaccination on the haemostatic system of T2DM patients with polyneuropathy with that of patients without polyneuropathy. COVID-19 vaccination had no impact on the haemostic system of patients with polyneuropathy. In people with T2DM without polyneuropathy D-Dimer levels significantly increased (p = 0.017); CTs were significantly shortened (p < 0.001), MCFs (p < 0.001) and alpha angles (p < 0.001) were significantly higher after vaccination (compared to the respective baseline levels). Moreover, platelet aggregation was slightly, but significantly, enhanced after vaccination. Amplitudes (p = 0.002) and Slopes (p = 0.041) were increased and Lag times were shortened (p < 0.001) after vaccination.”

Ad 6)

Participants in the COVAC-DM trial received their vaccination according to the Austrian vaccination plan and have therefore received two doses of the same vaccination (however due to the SMPCs the time span between the two doses differed). Thus, in the present study, homologous vaccination was applied. It has to be stated that a heterologous vaccination schedule has been shown to induce a significantly higher humoral response (Barocci et al. Evaluation of Two-Month Antibody Levels after Heterologous ChAdOx1-S/BNT162b2 Vaccination Compared to Homologous ChAdOx1-S or BNT162b2 Vaccination. Vaccines (Basel) 2022, 10, doi:10.3390/vaccines10040491.). Since their study also demonstrates that diabetes is not affecting the antibody titer, a heterologous vaccination schedule might also be well suited for people with T1DM or T2DM.

It is stated now in lines 320-324:

“Participants in our study have received two doses of the same vaccine. It has been shown that a heterologous vaccination schedule could induce a significantly higher humoral response in people with diabetes,[29] however, data on coagulation were not reported in this manuscript. Hence, we cannot exclude that a heterologues vaccination scheme might have a different impact on coagulation.”

Ad 7)

We completely agree with the reviewer that the moderate coagulation changes might be attributable to individuals’ own biological fluctuations. The coagulation values presented here are mainly derived from measurement performed in whole blood, leading to coefficients of variation greater than 10% (within-subject CVs). With the exception of MCF which has been shown to be fairly constant (6 %) (Jilma-Stohlawetz et al. Evaluation of between-, within-, and day-to-day variation of coagulation measured by rotational thrombelastometry (TEM). Scand j Clin Lab Invest 2017; 77: 651-657). For example, MCFs in T1DM patients were increased after vaccination by 11% (from 55 to 61 mm), suggesting an increase beyond physiological fluctuations. But nevertheless we speak in our manuscript only of subclinical alterations (calculated by the Friedman test) and not of clinically significant alterations. However, we have added a sentence in the discussion mentioning, that most of the alterations were within the normal range of the respective parameter (lines 266-267).

Ad 8)

We completely agree with the reviewer that inflammation leads to enhanced blood clotting. Unfortunately, the inflammation marker PAI-1 is not available. We, therefore, used high sensitivity CRP values to estimate possible coagulation activation after vaccination. We have added now the respective CRP values to Table 2 (T1DM patients) as well as to Table 3 (T2DM patients). We, however, did not see any increase of CRP values after vaccination. Thus, coagulation activation due to inflammation apparently plays a minor role in our study.

Ad 9)

Yes, the reviewer is completely right. It has been shown that platelet aggregation is strongly dependent on the diabetes duration. High platelet aggregation may become more pronounced with increasing diabetes duration caused by variations in hemorheological properties resulting in endothelial dysfunction (Yeom et al. Effect of diabetic duration on the hemorheological properties and the platelet aggregation in streptozotocin-induced diabetic rats. Sci Rep, DOI:10.1038/srep21913). We, therefore, compared the platelet aggregation value “Amplitude” of five patients with the shortest duration of diabetes with that of five patients with the longest duration of diabetes in both T1DM and T2DM groups. However, we saw virtually the same effect of vaccination in patients with long and with short diabetes duration in both T1DM and T2DM groups.

Ad 10)

The PPP was centrifuged at 2600 x g immediately before it was used for CAT measurements. Then, plasma was carefully aliquoted to the 96-well plate leaving 100 µl at the bottom of the vial.  We developed and validated these procedures years ago when we were studying the impact of microparticles on thrombin generation (Schweintzger et al. Thromb Res. 2011, Deutschmann et al. JPGN 2013). With our centrifugation scheme larger cell debris is removed, but the smaller fraction of microparticles remain in the sample. However, the method remains unaffected by microparticles when an excess of  4 µM phospholipids and 5 pM TF are added exogenously to the sample.

Ad 11)

Laboratory assays were performed in different batches. Intra- and inter-assay precision of the Atellica COAG 360 show coefficients of variations (CVs) < 5% for the present coagulation parameters (Hörber et al. Evaluation of the Atellica COAG 360 coagulation analyser in a central laboratory of a maximum care hospital. Int J Lab Hematol 2019; 00, 1-9.). Sysmex KX-21 derived parameters show CVs of 0.81 (HGB), 3.13 (PLT) (Fahres. Performance evaluation of two haematology analyzers: the Sysmex KX-21 and the Beckham Coulter Ac.T diff. Sysmex J Int 2001; 11). D-Dimer ELISA Kit: intra-assay reproducibility: CV < 10%, inter-assay reproducibility: CV < 12%. Thrombelastometry: Duplicate measurements showed that 23% of the CT and 31% of the CFT measurements had a coefficient of variation (CV) greater than 10%. The within-subject CV was 16% for the CT and 30% for the CFT. The MCF was fairly constant (6%). The between-subject CV was 6% for the CT and 20% for the CFT.(Jilma-Stohlawetz et al. Evaluation of between-, within-, and day-to-day variation of coagulation measured by rotational thrombelastometry (TEM). Scand j Clin Lab Invest 2017; 77: 651-657). Inter-assay variability of CAT is only 5% due calibration for each sample, and baseline, V1, and V2 of each patient were measured within one batch to minimize variability.

Given the laboratory variability, particularly for measurements performed in whole blood samples, we did not over-interpret our results. Although various coagulation values were altered significantly after vaccination (from a mathematical/statistical point of view), we speak only of subclinical alterations in our manuscript.

Reviewer 2 Report

I read with interest and I believe that the article can be published and of international interest especially on the topic considered. Allow me to add some suggestions:

- the limits of the study should be better explained

- the conclusions could also be extended regarding the fact that this risk does not compromise the safety of vaccination and therefore that there are no reasons to refuse it. doi: 10.3390 / vaccines9050500

Author Response

Response to Reviewer 2:

We thank the reviewer for the valuable comments helping to improve the quality of our manuscript!

Ad 1)

A further limitation of the study is that the vaccination-associated changes in several coagulation parameters were, although significant from a statistical/mathematical point of view, small without noted clinical significance. Thus, individuals’ own biological fluc-tuations together with laboratory variability might have influenced the data presented.

This is stated now in the discussion section, lines 340-343.

Ad 2)

As suggested by the Reviewer, conclusions are extended now. It is stated in lines 348-349: 

“The data presented herein support the safety profile of COVID-19 vaccination in people with diabetes, a population, in which the vaccination is highly recommended.[30]”

Round 2

Reviewer 1 Report

Thanks to the authors for the clarifications requested. However, I would like to suggest some more points and ask for clarification of some details:

1-In comment number 2, regarding the patients included, the authors comment that women taking oral contraceptives were not excluded from the study. Due to the large number of studies demonstrating the relationship between these treatments and thrombogenicity, I suggest indicating the number of patients taking oral contraceptives and including a study that compares the haemostatic profile of patients with this treatment versus those women not taking it, so as to reflect that the inclusion of women on oral contraceptive treatment is not altering the result obtained in the study.

2. I suggest including this additional information in the corresponding section on materials and method: Thrombin generation wavelengths were: 390 nm (excitation), 460 nm (emmission).

3- In comment 5, the authors explain that those patients with type II diabetes who were on ASA treatment had no significant differences between baseline and post-vaccination measurements, in contrast to those patients taking ASA. The authors believe that the fact that this difference was not observed is due precisely to the treatment with ASA, since taking ASA would be favouring the opposite effect and masking the shortening of CFT and an increase in MCF and alpha angle, as observed in patients without ASA?

On the oher hand, the authors included section 3.4 in the new submission; however, I believe that this should be explained in the discussion, commenting on how this might affect the observed results.

Author Response

Comments to Reviewer 1:

  • Ad 1) As stated by the reviewer, the inclusion of women on oral contraceptive treatment might have influenced the results obtained in the study. However, only one woman was on oral contraceptive treatment, so a sub-group analysis was not feasible. This is stated now in the Results section, line 177.
  • Ad 2) We have added now to the manuscript: “Thrombin generation wavelengths were: 390 nm (excitation), 460 nm (emission)”, lines 150-151.
  • Ad 3) As suggested by the reviewer, we state now in the manuscript that: ”Apparently, taking ASA masks the shortening of CFTs and the increases of MCF and alpha angle as observed in patients without ASA.”, lines 229-230.

As suggested by the reviewer, section 3.4 is removed now. It is stated now in the Discussion section: “COVID-19 vaccination had no impact on the haemostic system of patients with polyneuropathy. However, in people without polyneuropathy, D-Dimer levels were significantly increased, CTs were significantly shortened, MCF and alpha angles were significantly higher and platelet activation was slightly, but significantly, enhanced after vaccination. To our knowledge, the underlying mechanisms explaining this relative insusceptibility of people with polyneuropathy are not known today and should be clarified in a future study.” This is stated now in lines 336-342.